# Reformulation of Top-Selling Processed and Ultra-Processed Foods and Beverages in the Peruvian Food Supply after Front-of-Package Warning Label Policy

**DOI:** 10.3390/ijerph20010424

**Published:** 2022-12-27

**Authors:** Lorena Saavedra-Garcia, Mayra Meza-Hernández, Francisco Diez-Canseco, Lindsey Smith Taillie

**Affiliations:** 1CRONICAS Center of Excellence in Chronic Diseases, Universidad Peruana Cayetano Heredia, Lima 15074, Peru; 2Carolina Population Center, University of North Carolina at Chapel Hill, Chapel Hill, NC 27516, USA; 3Department of Nutrition, Gillings School of Global Public Health, University of North Carolina at Chapel Hill, Chapel Hill, NC 27516, USA

**Keywords:** food labeling, food policy, food supply, front-of-package labeling, Latin America, noncommunicable disease prevention, obesity prevention, sodium reduction, sugar policy

## Abstract

Front-of-package warning label (FOPWL) policies incentivize the food industry to reduce the content of regulated nutrients in products. We explored changes in the content of nutrients of concern (sugar, saturated fat, trans fat, and sodium) and the percentage of products in the Peruvian food supply that would carry a FOPWL before and after Peru’s implementation of FOPWLs. Longitudinal data on the top-selling foods and beverages (*n* = 94) were collected at three time points: three months before the implementation of the policy, four months after, and two years after. Using the nutritional information declared on products’ labels, we compared quantities of nutrients of concern and the percentage of foods that would carry a FOPWL at each time point. Between the first and the third data collection, a decrease in the median sugar content of beverages was observed (from 9.0 to 5.9 g/100 mL, *p* = 0.005), accompanied by an increase in the use of nonnutritive sweeteners. This change drove the reduction of the percentage of beverages that would carry a FOPWL (from 59 to 31%, *p* = 0.011). Among foods, decreases were observed in saturated fat (from 6.7 to 5.9 g/100 g, *p* = 0.002). The percentage of foods that would carry a FOPWL according to their nutritional profile declined from before to after implementation of the policy (from 82 to 62%, *p* < 0.001). The study shows that the industry reformulated products in Peru after implementation of its FOPWL policy.

## 1. Introduction

During the last few decades, the prevalence of obesity has grown quickly in low- and middle-income countries, becoming a pressing public health concern [1]. In Peru, the prevalence of excess weight among people older than 15 years increased from 53.3 to 62.5% between 2015 and 2020 [2]. Similarly, the prevalence of children between 5 and 9 years old with overweight and obesity was 19.4% in 2008 and increased to 37.4% ten years later [3]. This rise in the prevalence of obesity has been accompanied by an increase in ultra-processed food sales and intake. A decade ago, Peru had the lowest retail sales of ultra-processed foods per capita in Latin America, but between 2009 and 2019 it saw the highest increases in ultra-processed foods sales in the region [4]. Ultra-processed foods are often high in sugar, saturated fats, and sodium [5], and increased intake of these foods is associated with excess calorie intake, excess weight gain, increased cardio metabolic risk [6], and risk of noncommunicable diseases (NCDs) [7]. 

To prevent further increases in obesity and NCDs, Peru adopted a set of policies addressing various risk factors, including intake of ultra-processed foods. In 2013, the Peruvian Congress passed Law N°30021: Promotion of Healthy Eating for Children and Adolescents [8]. In addition to other policies, such as eliminating trans fats [9] and modifying an existing tax on sugar-sweetened beverages (SSBs) [10], the law established the mandatory use of front-of-packaging warning labels (FOPWLs), for packaged processed foods and beverages (including ultra-processed products) that exceed nutritional thresholds, in the form of black octagons indicating that the products are “high in” sugar, saturated fats, or sodium or contain trans fats [8]. Despite strong opposition [11], the policy was implemented in two phases (the first starting in June 2019 and the second in September 2021), with thresholds becoming more restrictive in the second phase [12]. 

One major question is how the Peruvian food supply changed after the implementation of this policy. In addition to informing consumers, FOPWLs incentivize the food industry to reduce the content of these nutrients of concern in their offerings, whether through product reformulation, new product entry, or removal of products [13]. For instance, Chile [14,15] and Australia [16] have seen the reformulation of processed and ultra-processed foods in response to the implementation of front-of-package labeling efforts. This reformulation resulted in decreasing the content of nutrients of concern by reducing them and/or replacing them with other ingredients [17]. Reformulation allows companies to avoid the placement of FOPWLs on their products, thereby reducing the potential negative impact on their sales. For example, after the implementation of Chile’s FOPWL policy, researchers found that, in conjunction with a decrease in sugar, the use of nonnutritive sweeteners (NNS) increased in products such as nonalcoholic beverages, cereals, dairy products, and desserts [18,19]. However, it is not clear whether the food supply in Peru responded similarly to its FOPWL regulation. Understanding the presence and extent of reformulation is important because, while reduction of nutrients like sodium and sugar can be beneficial for health [20], these benefits may be offset depending on what ingredients and additives are used to replace them [17].

The objective of this study was to understand changes in the content of nutrients of concern (sugar, sodium, saturated fat, and trans-fat) and to determine the prevalence of products that would be considered “high-in” nutrients of concern (i.e., those that would receive the FOPWL) among the top-selling processed foods and beverages in the Peruvian food supply before and after implementation of the mandatory FOPWL policy. This study allowed us to capture rapid changes in reformulation of products widely offered to and consumed by the majority of the population in Peru.

## 2. Materials and Methods

### 2.1. Study Design 

This was a longitudinal study of the top-selling processed foods and beverages in the Peruvian market. In brief, nutritional information from the same products was collected at three time points: in March 2019, three months prior to the implementation of the first phase of the FOPWL policy; in October 2019, four months after implementation; and in May 2021, two years after implementation. 

### 2.2. Setting

Data collection of the top-selling food and beverages was conducted in Lima, Peru’s capital and the largest, most commercial city in the country. For the pre-implementation measurements, in March 2019 data from top-selling products were obtained from three different locations of a large national chain of Peruvian supermarkets. As the intent of the study was to capture a variety of products, each of the selected three supermarkets targeted one of three different socioeconomic sectors (high, medium, and low) thereby opening the possibility to find most of the selected products. 

For the postimplementation measurements, the same three supermarkets were visited in October 2019 and May 2021, as well as one *bodega* (small grocery store) and one convenience store. The *bodega* and convenience store were added because three of the top-selling products were not found in the supermarkets during the follow-up visits but were commonly sold in small retail stores. During the third data collection, product availability in supermarkets was also reduced because of the COVID-19 lockdown. 

### 2.3. Sampling Methods

To select the top-selling foods and beverages that were collected at the three time points, we used 2018 data on Peru from Euromonitor [21] and Kantar World Panel, two international market research companies that provide information about sales and purchases of foods and beverages. To begin selecting the top-selling products for the study, we had to determine meaningful categories of foods and beverages. These categories included products that were likely to be regulated by the FOPWL policy (for instance, single-ingredient foods like oil or honey as well as unprocessed foods are exempt from FOPWL regulations and therefore were excluded from this study) and were potential candidates for reformulation according to previous studies in Latin America [22]. Based on those principles, 13 beverage categories and 35 food categories were selected from 2018 Euromonitor data. Then the top brands in each of the 48 categories of interest were selected. These brands represented more than 60% of the total retail sales in each category, except for cookies, chocolate-coated biscuits, filled biscuits, plain biscuits, and savory biscuits, in which “other” brands have a larger percentage of market sales. For those categories, the brands selected represented 40% of the sales.

Up to four brands were selected in each food or beverage Euromonitor category; a total of 28 brands were selected for beverages and 70 brands for foods. 

Because the Euromonitor data contain only brands and not the names of individual products, and there are many products within a brand in some cases, to determine the most relevant products for inclusion our team used 2018 Kantar World Panel data, collected before the FOPWL implementation, which includes product and purchase information from 3000 urban households. Within each of the 98 brands, we selected the most-purchased products (specific products of specific flavors and sizes). Alternative sizes or flavors of the same product were identified for situations where the selected product may not have been available. Finally, in each of the 48 Euromonitor categories, from one to four of the top-selling products were selected. The list of products comprised 71 beverages and 99 foods (*n* = 170), including 105 main products and 65 alternatives (i.e., different flavors), in case the fieldwork team did not find the 105 main products.

For analysis, the 48 chosen categories were collapsed by trained nutritionists to follow a categorization scheme common for Peruvian foods. The beverages were organized into nine categories and the foods into six (see Appendix A).

### 2.4. Data Collection and Entry

Top-selling products were obtained mainly from three supermarkets by nutritionists in charge of the search. If a product was not found in the largest supermarket, then a separate visit was made to the second one, and finally to the third. If a product was not located after visiting all of the supermarkets twice, the product was considered to be discontinued. Products were identified by checking the brand, manufacturer, size, and flavor as listed on the product; because barcodes could change over time for the same product, it was acceptable if the barcode was not the same. If the same size of the product was not available on the day of the supermarket visit, the closest size to the original was purchased and considered a match; it was assumed that reformulation was the same across all sizes of a product. During the third collection in May 2021, after performing the search in the three supermarkets, one *bodega* and one convenience store were visited to complete the data collection. Those additional retail locations were considered as the foods offered were reduced during the COVID-19 pandemic.

During the data collection in March 2019, before implementation of FOPWLs, a total of 32 beverages and 73 foods from the 170 products originally selected were photographed. During the second data collection in October 2019, 32 beverages and 72 foods previously collected were located and photographed. During the third data collection in May 2021, 31 beverages and 69 foods previously collected were found and photographed. Only 94 out of these 100 products were included in the final analysis because 3 products did not declare nutritional information and 3 products did not declare the nutrients of concern, so the analysis on them could not be performed (Figure 1).

During each data collection, photographs of all sides of the packages were taken, capturing nutrition facts panels and ingredient lists that later were entered into a REDCap [23] database hosted at the University of North Carolina at Chapel Hill. Nutrient quantities were programmatically standardized to the content per 100 g or 100 mL of the product and then reviewed by trained nutritionists. To include as many products as possible, the 12 products (five beverages and seven foods) requiring preparation to be consumed were “prepared” according to the packages’ instructions, as the Peruvian norm indicates that the appearance of FOPWLs should be based on nutrient values of products as consumed. For those products, the nutrient values of ingredients added for product preparation were based on the USDA Food and Nutrient Database for Dietary Studies [24].

To ensure data quality, all information from the nutrition facts and ingredient lists entered in the database was then reviewed and compared with the photos, and any mistakes were corrected. 

### 2.5. Outcomes

This study had three outcomes: 

the content of nutrients of concern (grams of total sugar, saturated fats, trans fats, and sodium) per 100 g or 100 mL of products as consumed;the percentage of products containing NNS, including polyols (see Appendix A), and monosodium glutamate (MSG), ingredients used to replace sugar and salt, respectively;the percentage of processed and ultra-processed products that would carry FOPWLs according to the parameters of the first phase of the FOPWL policy (Table 1). 

### 2.6. Statistical Analysis

The median and interquartile range (IQR) were used to describe quantities of nutrients of concern per 100 g or 100 mL overall and by food and beverage category for all three data collection points. To determine differences in nutrient content by groups (food and beverages) and by categories across time, we used the Friedman test as a nonparametric test for three groups of paired data. Statistical significance was considered at alpha = 0.05. If significance was found, we used the post hoc paired comparison Wilcoxon signed-rank test. As not all companies included the nutritional facts panel on the label, products not declaring a particular nutritional value were not considered for analysis of that particular nutritional information (see Appendix A). 

In addition, the percentage of products that contained NNS and MSG was calculated. To determine differences by beverage and food categories across time, we used the Cochran test for three groups of paired data. If significance was found, the post hoc paired comparison McNemar test was applied. 

Finally, the percentage of products “high-in” for each nutrient of concern or containing trans fats was determined and compared using McNemar’s test for paired data. Analyses were performed in SAS v9.4.

## 3. Results

A total of 94 products were included in the analyses, 30.9% (*n* = 29) from the beverages group and 69.1% (*n* = 65) from the foods group. Table 2 shows the median content of the four nutrients (i.e., sugar, saturated fats, trans fats, and sodium) regulated by the Peruvian FOPWL policy at each time point. Compared to the pre-policy collection, decreases in sugar content were observed in beverages (*p* = 0.005) in the post-FOWL period, but there were no corresponding changes in foods. 

Regarding fats, we did not observe changes among beverages, but for foods, decreases were observed in saturated fat (*p* = 0.002) and trans fats (*p* = 0.021). Interestingly, most changes were observed between the first and second collections (i.e., between pre-policy and the first collection in the post-policy period), and no significant changes were observed between the second and third collections (i.e., four months and two years after FOPWL implementation). In addition, no statistically significant changes were observed for sodium. MSG was not used in beverages. In foods, the percentage of products using it was 17% (*n* = 11), which remained consistent across the three collections. 

Among beverages, the decline in sugar content was accompanied by an increase in the use of NNS (Figure 2), which increased in prevalence from 34.5% of beverages before the law to 62.1% in the third collection. For foods, the prevalence of NNS increased from 15.4% in the first collection to 20% in the third.

Regarding products that would carry a FOPWL using the parameters of the first implementation phase, at baseline 59% of beverages would display an octagon, but at the third collection (two years after implementation), the percentage dropped to 31% (*p* = 0.011). The change was driven by the decrease of beverages “high-in sugar” (*p* = 0.005), and no beverages carry a “high-in saturated fats” octagon or a “high-in sodium” octagon (Figure 3). With regard to food products, the percentage of the sample that would carry any FOPWL label decreased from 82% to 62% (*p* < 0.001). These decreases were observed in the prevalence of products carrying a label for each of the four nutrients of concern, although the largest changes were observed in products that would carry a “high-in sugar” (*p* = 0.008) and “contains trans fats” (*p* = 0.008) warning label. The percentage of foods that would carry a trans fat or sodium warning label was low (under 15%) across all three time periods, including the period before regulation. 

## 4. Discussion

This is the first study to evaluate changes in the prevalence of FOPWLs and the content of nutrients of concern in top-selling packaged beverages and foods in the Peruvian market after the implementation of Peru’s warning label policy. According to our results, two years after policy implementation, the prevalence of any beverage carrying FOPWLs dropped by 28 percentage points, and the prevalence of any foods carrying FOPWLs dropped by 20 percentage points. In 2021, only about a third of beverages had a FOPWL, whereas more than half (62%) of foods still should carry a FOPWL according to the nutrient thresholds.

We observed reductions in nutrients of concern after policy implementation, but they were different for beverages and foods (see Table 2). Sugar content declined in beverages, and in parallel, the use of NNS increased. In contrast, there were no significant reductions in the sugar content of foods, but we observed reductions in the number of foods carrying a “high-in sugar” octagon (see Figure 3), which could be mean it was feasible to avoid the octagon even with small reductions in sugar content. In foods, there were reductions in saturated and trans fats. Interestingly, even prior to the warning-label law, the percentage of products containing trans fats was low.

Like other FOPWL systems, Peru’s warning-label system was designed to encourage reformulation such that companies would reduce nutrients of concern below the thresholds to avoid having to put warnings on their products. Other countries have implemented similar warning labels, but few studies regarding the impact of reformulation have been developed. Studies evaluating Chile’s FOPWL and advertising policy found similar results as ours, showing that the nutrient most reduced is sugar. Compared with the other nutrients of concern, these studies found that six out of 16 food and beverage categories reduced the proportion of products “high-in” sugar, followed by sodium in five categories, and saturated fats in only one category [15]. In contrast to our study, which observed decreases in the percentage of food products carrying sugar FOPWLs but not in the actual sugar content, the Chilean research found decreases in both sugar content and products carrying the sugar octagon in some food categories. It is possible that because of the limited sample size in this study (as we only considered some food categories and products), we did not observe significant changes in the sugar content. The increased use of NNS in food products suggests that a proportion of sugar may be being replaced. 

The Chilean studies found that replacing sugar with NNS facilitated the reformulation of sugar content, which was similar to our study’s findings. For instance, Chilean researchers found that the use of NNS in beverages increased from 72.0 to 82.6% (*p* < 0.001) after the implementation of mandatory FOPWLs [19], whereas in Peru, the use of NNS increased from 34.5 to 62.1%. The magnitude of the increase in NNS use Chilean beverages is lower than in Peruvian beverages (10 percentage points versus 28 percentage points, respectively). This larger increase in NNS use in Peru may be due to the fact that, relative to Peru, a much higher proportion of Chilean beverage products contained NNS prior to implementation of the FOPWL policy [25]. Substituting sugar for NNS is an expected strategy, especially in beverages where NNS are only used to add sweetness; this could be more difficult in food categories where sugar provides other properties (e.g., texture and bulking) [26,27]. It is also important to point out that the observed decreases in sugar content in beverages could have been driven jointly by the FOPWL policy and the Peruvian SSB tax, which entered into force only one year before the warning-label law [10]. It has been reported that SSB taxes could lead to reformulation, as happened in the United Kingdom [28]. 

In addition to changes in sugar content in beverages, we observed small but significant reductions of saturated fats and the elimination of trans fats in foods, although not enough to show a change in the percentage of products with octagons. The decrease in the amount of saturated fats in products may have been small because reductions in saturated fat create technical difficulties due to their high oxidative stability and melting points, although it has been suggested that decreases could be achievable with technological treatments such as oil interesterification for baked goods [27]. Still, the Peruvian results on saturated fat were consistent with the Chilean, which showed small changes in saturated fat food categories after the implementation of the first phase of FOPWL policy [15]. On the other hand, the percentage of products containing trans fats was small since the first collection (*n* = 10), and in the last collection, only six products continued to contain trans fats. This shows that the industry is aligning with the current Peruvian regulation implemented in 2016, which established the gradual reduction of trans fats and finally prohibited the use of trans fats as of July 2021 [9].

In contrast to the other nutrients of concern regulated by Peru’s warning-label policy, the content of sodium did not change in beverages or foods, and the number of products carrying a “high-in sodium” octagon was low. Even before the law’s implementation, most top-selling foods and all beverages had sodium levels below the cutoff points of the first phase of the Peruvian policy. These preexisting low sodium levels could be a spillover effect from other Latin American countries that established sodium targets for processed foods in previous years [29,30]. For example, the sodium threshold implemented for Chile’s FOPWL in 2016 was exactly the same implemented in 2019 in Peru. In the process of reformulating products for Chile, transnational companies may have also reformulated products for the Peruvian market, so that by the year of implementation, the sodium content in Peruvian products was relatively low compared with the thresholds, and the industry did not need to additionally reformulate to avoid carrying the sodium octagon. It may be possible to find reductions closer to or after the second phase (implemented in September 2021), as it established stricter thresholds for sodium in food products that could incentivize reformulation.

Given that the Peruvian law required increasingly more strict nutrient thresholds over two phases, we expected to observe continuing declines in the median content of nutrients of concern and FOPWLs over time. Specifically, we expected that nutrient concentrations of products from the third collection would be lower than nutrient concentrations of products from the second collection because the third collection was carried out only four months before the second phase was implemented. Yet we found no evidence of reformulation beyond what initially occurred after implementation of the first phase. One possibility is that companies simply had not implemented any additional reformulations yet. It is also possible that the industry found that more reformulated products would not be accepted by consumers because of changes in the taste, structure, or mouthfeel of the product, which indicates that reformulation depends not only on what is technologically feasible but also on public acceptance [26]. A third possibility is that we missed any additional reformulations that occurred in the food supply because of the small sample size.

Our results are useful for policy makers to evaluate, complement, and strengthen the current regulation in Peru. For instance, other countries such as Mexico and Argentina have implemented NNS warning labels as part of their FOPWL systems [31,32], because of health concerns related to ultra-processed foods as well as the association of some NNS with metabolic disruption [33]. Our results show that without a warning for NNS, warning-label policies that include sugar will likely lead to an increase in NNS. Peruvian policy makers could consider introducing an NNS warning to avoid further increases in the prevalence of these ingredients. 

We acknowledge the limits of the study. First, in order to conduct rapid data collection over more frequent time periods, only small samples of products were used, and only some categories of food and beverages were selected. These small sample sizes precluded our ability to examine changes in nutrients in food and beverage subcategories. However the products selected represented the top-selling items in Peruvian urban households, suggesting that they are the most consumed products in the country. Second, products included in the analysis were those available at the points of sales visited in Lima for each collection, so it is possible that products not found were offered in other retail locations. Third, because in Peru it is not mandatory to declare nutrition facts, we excluded six products that did not provide nutritional information and missed nutrient content data from some products in the final sample. Fortunately, most products in the sample declared nutrients of concern. Future studies to evaluate changes in nutrient composition within the broader food supply may face challenges because of the lack of nutrition facts data. 

This study also had several strengths. A major strength is the longitudinal design of the study; beverages and foods were matched according to barcode and product features to ensure comparison across time. Our study also captured data from three time points, including the last one close to the second phase of FOWPL implementation, and considered some replacement ingredients for nutrients of concern, which gives us a better overview of the implementation of FOPWL.

## 5. Conclusions

The study shows that, after the implementation of FOPWL in Peru, companies reformulated foods and beverages to decrease nutrients of concern, resulting in a reduction in the prevalence of products required to carry the FOPWL. These changes occurred mostly by replacing sugars with NNS, especially in beverages, and reducing saturated and trans fats in foods. Further studies considering a larger sample of products from the Peruvian market should be conducted. Finally, to help consumers make more informed food choices, policy makers should strengthen the current Peruvian norm by considering including NNS warning labels and making the nutrition facts panel mandatory on products’ labels to complement the information provided by the FOPWL.

## Figures and Tables

**Figure 1 ijerph-20-00424-f001:**
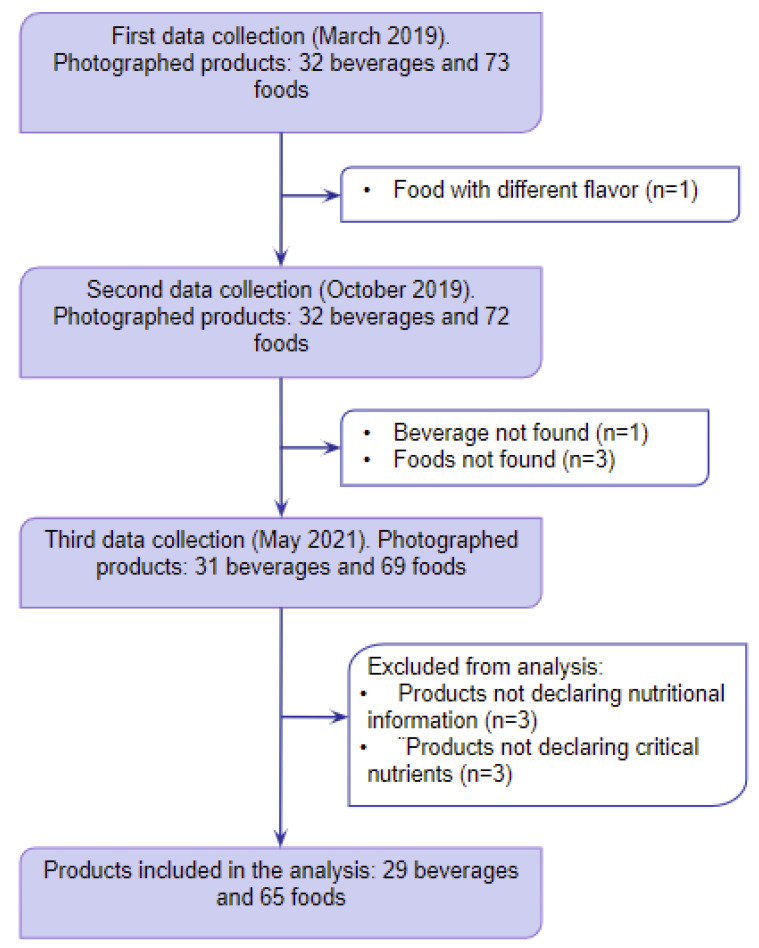
Sample selection process.

**Figure 2 ijerph-20-00424-f002:**
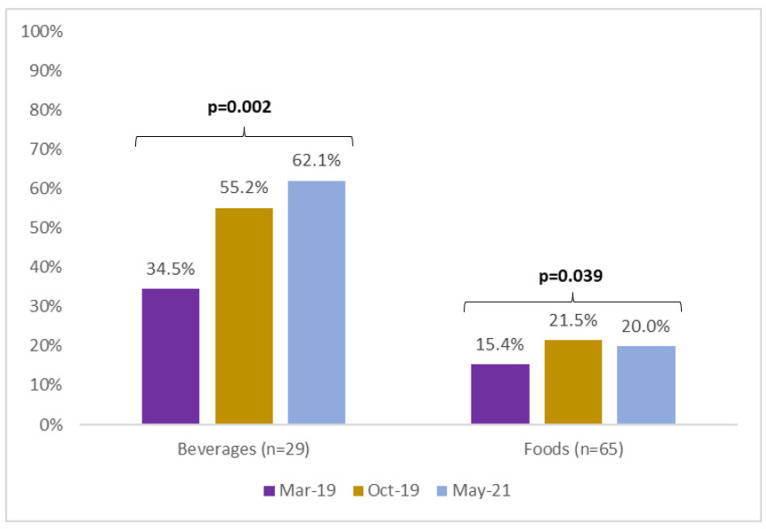
Percentage of beverages and foods using nonnutritive sweeteners (NNS) across time. Comparison of percentages of products using NNS was made using the Cochran test for three groups of paired data. Bold values represent *p* < 0.05.

**Figure 3 ijerph-20-00424-f003:**
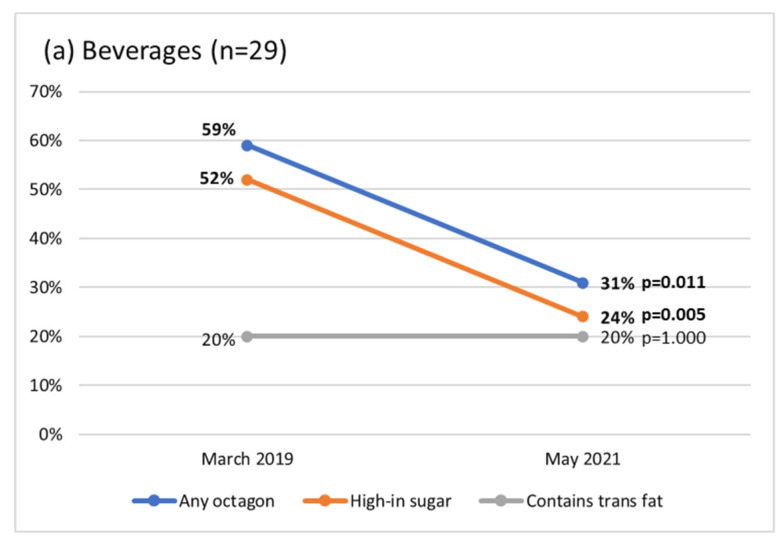
Percentage of products that would carry front-of package warning labels before and after the policy implementation according to the first-phase nutritional thresholds. Comparisons were made using the Cochran test. Bold values represent *p* < 0.05.

**Table 1 ijerph-20-00424-t001:** Technical parameters and entry into force of the Promotion of Healthy Eating for Children and Adolescents law.

Nutrient of Concern	Product	Term of Entry into Force	
		First Phase (June 2019)	Second Phase (September 2021)
Total sugars	Solids	Greater than or equal to 22.5 g/100 g	Greater than or equal to 10 g/100 g
	Beverages	Greater than or equal to 6 g/100 mL	Greater than or equal to 5 g/100 mL
Saturated fats	Solids	Greater than or equal to 6 g/100 g	Greater than or equal to 4 g/100 g
	Beverages	Greater than or equal to 3 g/100 mL	Greater than or equal to 3 g/100 mL
Sodium	Solids	Greater than or equal to 800 mg/100 g	Greater than or equal to 400 mg/100 mL
	Beverages	Greater than or equal to 100 mg/100 mL	Greater than or equal to 100 mg/100 mL
Trans fats	Solids and beverages	Contains trans fats	Added trans fats prohibited [10]

**Table 2 ijerph-20-00424-t002:** Nutrients of concern per 100 g/mL in processed and ultra-processed products across time ^1,2^.

Nutrient	Period	Beverages (*n* = 29)	Foods (*n* = 65)
Sugar (g)	*n* (%) *	29 (100)	64 (98.5)
	March 2019	9.0 (4.7) ^a^	11.4 (31.9)
	October 2019	5.9 (3.8) ^b^	16.4 (26.3)
	May 2021	5.9 (3.1) ^b^	15.6 (28.0)
Saturated fat (g)	*n* (%) *	13 (44.8)	62 (95.4)
	March 2019	1.0 (1.0)	6.7 (10.6) ^a^
	October 2019	1.0 (1.0)	5.4 (11.6) ^b^
	May 2021	1.0 (1.0)	5.9 (11.6) ^b^
Trans fats (g)	*n* (%) *	10 (34.5)	61 (93.8)
	March 2019	0 (0)	0 (0) ^a,†^
	October 2019	0 (0)	0 (0) ^b^
	May 2021	0 (0)	0 (0) ^b^
Sodium (mg)	*n* (%) *	29 (100)	65 (100)
	March 2019	16.0 (37.8)	339.0 (451.0)
	October 2019	16.0 (41.0)	343.9 (430.0)
	May 2021	25.5 (42.0)	339.0 (478.0)

^1^ Data expressed through medians and interquartile ranges (IQR); parenthetical values represent the IQR. ^2^ Letters ^a^ and ^b^ indicate statistically significant differences. * Refers to the number (and percentage) of products available for that nutrient within the food or beverage category. ^†^ Minimum–maximum value: 0.0–7.7.

## Data Availability

The data that support the findings of this study are available from Kantar/Europanel. Restrictions apply to the availability of these data, which were used under license for this study. Interested parties can contact understand@europanel.com to inquire about accessing this proprietary data.

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
