# Peer review of "Reformulation of Top-Selling Processed and Ultra-Processed Foods and Beverages in the Peruvian Food Supply after Front-of-Package Warning Label Policy"

_ijerph, 2022, doi:10.3390/ijerph20010424_

Round 1
Reviewer 1 Report
Dear authors,
Please revise the manuscript accordingly.
Thanks,
Reviewer.

Author Response
We wish to express our gratitude to the reviewer for their time invested in reading our manuscript and their enriching comments which will result in an improved version of the manuscript.
Please find attached our point-by-point response to all the comments provided. All changes have been included in the manuscript using track changes.
The authors

Reviewer 2 Report
GENERAL COMMENTS
Dear authors,
Congratulations on your manuscript.
The manuscript well written, easy to follow, and looks sound.
I have some minor questions.
Kind Regards
Reviewer
SPECIFIC COMMENTS
Lines 18 and 87: In line 18 4 months are mentioned, but in line 87 3 months?
Line 96: What were the criteria used to classify the supermarkets in the three socioeconomic sectors? It would be important to clarify this point.
Author Response
We wish to express our gratitude to the reviewer for their time invested in reading our manuscript and thankful for their enriching comments which will result in an improved version of the manuscript.
Please find attached our point-by-point response to all the comments provided. All changes have been included in the manuscript using track changes.

Round 2
Reviewer 1 Report
Dear authors,
Please consider revising accordingly.
Reviewer.

Author Response
We appreciate the feedback provided by the reviewer and we are grateful for the opportunity to submit a revised version of our manuscript.
Below you will find a point-by-point response to each comment.
-
1. Consider changing to (i.e., sugar.....)
Response: Thank you for the suggestion. We have modified as suggested.
2. Change 6 to six.
Response: We have corrected the number.
3. Remove additional space(s).
Response: We have rectified this.
4. Consider rephrasing lines 374-376.
Response Thanks for your suggestion. We have now clarified the statement.